# Adolescents and age of consent to HIV testing: an updated review of national policies in sub-Saharan Africa

Magdalena Barr-DiChiara [iD],[1] Mandikudza Tembo,[2,3] Lisa Harrison,[4] Caitlin Quinn,[1] Wole Ameyan,[1] Keith Sabin,[5] Bhavin Jani,[6] Muhammad S Jamil,[1] Rachel Baggaley,[1] Cheryl Johnson[1]

[1]Global HIV, Hepatitis and STI Programmes, World Health Organization, Geneva, Switzerland
[2]Department of Infectious Disease Epidemiology, London School of Hygiene and Tropical Medicine, London, UK
[3]Biomedical Research and Training Institute, Harare, Zimbabwe
[4]South West Screening and Immunisations, Public Health England, London, UK
[5]Strategic Information and Evaluation, UNAIDS, Geneva, Switzerland
[6]World Health Organization, Dar es Salaam, Tanzania

**Correspondence to**
Magdalena Barr-DiChiara;
m.dichiara@gmail.com

## ABSTRACT

**Objectives** In sub-Saharan Africa (SSA) where HIV burden is highest, access to testing, a key entry point for prevention and treatment, remains low for adolescents (aged 10–19). Access may be hampered by policies requiring parental consent for adolescents to receive HIV testing services (HTS). In 2013, the WHO recommended countries to review HTS age of consent policies. Here, we investigate country progress and policies on age of consent for HIV testing.

**Design** Comprehensive policy review.

**Data sources** Policies addressing HTS were obtained through searching WHO repositories and governmental and non-governmental websites and consulting country and regional experts.

**Eligibility criteria** HTS policies published by SSA governments before 2019 that included age of consent.

**Data extraction and synthesis** Data were extracted on HTS age of consent including exceptions based on risk and maturity. Descriptive analyses of included policies were disaggregated by Eastern and Southern Africa (ESA) and Western and Central Africa (WCA) subregions.

**Results** Thirty-nine policies were reviewed, 38 were eligible; 19/38 (50%) permitted HTS for adolescents ≤16 years old without parental consent. Of these, six allowed HTS at ≥12 years old, two at ≥13, two at ≥14, five at ≥15 and four at ≥16. In ESA, 71% (n=15/21) allowed those of ≤16 years old to access HTS, while only 24% (n=6/25) of WCA countries allowed the same. Maturity exceptions including marriage, sexual activity, pregnancy or key population were identified in 18 policies. In 2019, 63% (n=19/30) of policies with clear age-based criteria allowed adolescents of 12–16 years old to access HIV testing without parental consent, an increase from 37% (n=14/38) in 2013.

**Conclusions** While many countries in SSA have revised their HTS policies, many do not specify age of consent. Revision of SSA consent to HTS policies, particularly in WCA, remains a priority to achieve the 2025 goal of 95% of people with HIV knowing their status.

## BACKGROUND

In the past decade, substantial but unequal progress has been made towards the global '90-90-90' fast-track goals: in 2019, an estimated 81% of people living with HIV (PLHIV)

### Strengths and limitations of this study

► This study provides a comprehensive review of national age of consent to HIV testing policies in sub-Saharan Africa.
► This review included national policy documents but did not assess national laws, which contribute to a potentially limiting operational context to better understand guidance, which informs routine implementation.
► Due to varied document formats, the reviewers employed a non-automatised data extraction that was complemented by using key words to maintain repeatability.
► Despite a robust search approach, some policy documents may not have been identified.

know their status, 82% of aware PLHIV are on antiretroviral therapy (ART) and 88% of PLHIV on ART are achieving viral suppression.[1] Despite successes, nearly 19% of the 38 million PLHIV remain undiagnosed and there are an estimated 1.7 million new HIV infections annually.[2]

Globally, three-quarters (approximately 130 000) of the new 170 000 annual HIV infections among adolescents (aged 10–19 years) are in girls.[3] In sub-Saharan Africa (SSA), home to nearly 90% of all adolescents and young PLHIV, adolescent girls and young women aged 15–24 years make up one-quarter (about 330 000) of the new HIV infections although they represent just 10% of the population.[2] The number of young people in SSA is rapidly increasing, and it is estimated that there will be almost 500 million 15–24 year olds in the region by 2055.[2] Recent projections point to declines in HIV infections among young people in SSA but not enough to reach global targets for low HIV incidence by 2030.[2 4]

Despite the high burden and incidence and growing population, access to and uptake of

BMJ

HIV testing services (HTS), a pathway to essential prevention and treatment, remains remarkably low among adolescents.[5 6] In Eastern and Southern Africa (ESA), it is estimated that only 27% of adolescent girls and 16% of adolescent boys aged 15–19 years old received an HIV test in 2019, and lower uptake was observed in Western and Central Africa (WCA).[3] Access may be hampered by policies requiring parental consent for adolescents to receive HTS.[7–9]

Adolescence is a time of rapid physical, sexual and cognitive development. Health services for adolescents, including HTS, should be delivered through youth-friendly approaches that reflect their diverse and evolving needs.[10] Social determinants, such as wealth and gender inequality as well as access to health and education services, impact adolescent health behaviour and outcomes in the near term, as they transition into adulthood and throughout the life course.[11–14]

Particularly underserved are adolescents and young people from key populations: men who have sex with men, transgender people, young people who sell sex[i] (people 10–24 years of age, including children 10–17 years old who are sexually exploited and adults 18–24 years old who are sex workers), people who inject drugs and people in prisons or closed settings. Key populations, especially those of younger ages, are criminalised and face stigma and discrimination from community members and providers.[15–17]

Adolescents at high ongoing risk who do not know their HIV status can benefit from HTS, prevention or treatment, but access to these services is limited by policies requiring authorisation from a parent or guardian.[7–9 18] While parental support can have a positive effect on adolescent's sexual health decision-making, for many adolescents who fear disclosing their sexual and other risk behaviours to their parents, those who do not have positive parental relationships or those who live in child-headed households, parental consent requirements pose a major barrier to HTS.[19] Empowering adolescents to freely access HTS and to choose how they want to test, including to use HIV self-testing (HIVST), supports equitable service delivery and the right to access health services.[20–22] Earlier analysis of the relationship between HTS coverage of young people in SSA and age of consent policies showed that allowing free adolescent access was associated with testing coverage of 14.0 percentage points

higher for girls and 6.9 percentage points for boys.[8] In 2013, WHO guidance on HIV and adolescents encouraged countries to consider revising parental consent requirements to reduce barriers to HTS.[23] Since that time, the WHO has recommended HIVST, a convenient and private option, which has been demonstrated highly acceptable for adolescents in SSA.[24 25] Countries in SSA have moved to rapidly introduce and scale up differentiated HIVST approaches that link people to diagnosis, treatment and care. Recent findings show that when given the option of facility-based HTS or HIVST, adolescents and young people choose HIVST over conventional modalities and are more likely to test and share tests with peers and partners when using HIVST.[26 27]

Here, we review policies in force following the WHO 2013 guidelines to evaluate country progress and uptake of WHO guidance on age of consent for HIV testing, including the age of consent for accessing HIVST.[7 23 28]

## METHODS

### Search strategy

We carried out a search of national policy documents related to HTS through searching WHO repositories, governmental and non-governmental websites and search engines and contacting WHO country contacts and regional experts through April 2018 and updated our database through August 2020 (online supplemental file 1).[29]

### Inclusion criteria

All national policy documents pertaining to HTS were included: national HIV guidelines and strategic plans covering HIV and integrated guidelines on the prevention, treatment and care of HIV. For inclusion in the review, the policy must have included adolescent consent to HIV testing and be published by a WHO member state in SSA, of which there were 49. The policy language was not restricted, but in policies other than English, extraction of age of consent was prioritised. The date of publication was not restricted, and when more than one policy were identified for a country, the most current publication was included. When necessary, we contacted government officials, regional technical advisors and other key experts to gather and/or validate information.

### Data extraction

Policy data were extracted using a standardised Excel-based form (see online supplemental file 2, policy data extraction tool). Information extracted included age of consent to HIV testing and/or self-testing as well as exceptions, including but not limited to individual maturity, key population status, emancipation, parenthood, marital status, sexual activity and year of publication. HIV burden for included countries was classified as high ($\geq 5\%$) or low ($<5\%$) and collected from the Global AIDS Monitoring tool.[1 18] In addition to the policy assessment by the review team, national age of consent to HTS was

---

[i]While young adults who sell sex are considered sex workers, multiple international conventions describe the participation of those under 18 years of age in selling sex as a contravention of human rights law, and those who do so are considered sexually exploited. United Nations. United Nations Convention on the Rights of the Child (Article 1). U.N. Doc. A/Res/44/25, 1989 (http://www.ohchr.org/EN/ProfessionalInterest/Pages/CRC.aspx United Nations. United Nations protocol to prevent, suppress, and punish trafficking in persons, especially women and children. Supplementing the United Nations convention against transnational organised crime. Article 3 (a-d); 2001 (A/45/49, https://www.unodc.org/documents/treaties/UNTOC/Publications/TOCConvention/TOCebook-e.pdf.

**Table 1**  Definitions for country policies to age of consent to HIV testing

| Policy category | Definition |
| --- | --- |
| No mention (NM) | Policy does not mention adolescents or minors or age of consent for HTS |
| Minor not defined (MND) | Age of minor not defined |
| Parental consent required (PCR) | Age of consent to HIV testing is 18 years or above. Ability to consent is defined by age |
| Some adolescents may access without parental consent (SAC) | Age of consent to HIV testing is below 18 years. Minimum age to consent for HTS varies by country. Ability to consent is defined by age |
| No minimum age limit (NA) | There is no minimum age limit for independent consent to HIV testing |
| Mature minor exceptions (MME) | Minors classified as mature may access HIV testing without parental consent. Criteria for exceptions to minimum age of consent may include the following:<br>► Shows maturity and understanding of the process and potential results (healthcare provider discretion).<br>► At risk of contracting HIV or sexually active.<br>► Symptomatic.<br>► Pregnant.<br>► Parent (adolescent is already a parent).<br>► Head of household.<br>► Married.<br>► Child engaged in commercial sex work.<br>► Street child.<br>MME are not mutually exclusive of other approaches including no adolescent access, some adolescent access or no age limit |

HTS, HIV testing services.

also obtained from the National Commitments and Policy Instrument (NCPI) database, a global survey of HIV laws and policies.[28] Policies in languages other than English were reviewed with the aid of the translation software Google Translate.

Two authors (MB-D and MT) extracted information from policy documents independently.

Key word searching for terms including adolescent, child, minor, parent and consent was used to identify content addressing age of consent with full-text review completed on the sections identified through this process. Disagreements between reviewers were resolved by discussion and consultation with a senior author (CJ).

### Analysis

Countries were classified according to two subregions: ESA and WCA.[ii] Policy data were then categorised using previously developed definitions and categories (table 1): no mention, no adolescent access without parental consent, some adolescents may access without parental consent and no minimum age limit.[8]

We conducted a descriptive analysis, complementing 2019 policy data with results reported in NCPI. We assessed country policy changes since 2013, including variation in age of consent to HIV testing and maturity exceptions, by comparing our results to the previous WHO review on age of consent.[17]

---

[ii]We present data in two subregions, ESA and WCA, which closely align with UNAIDS regions. We include Djibouti and Somalia, which are not in these UNAIDS regions. We do not include Algeria, which is traditionally North Africa, though it sits in the WHO African region.

### Patient and public involvement

Patients and/or the public were not involved in the design, conduct or reporting of this research.

### RESULTS

A total of 39 national policies published from 2005 to 2019 were identified among SSA countries; for ten countries, no policy was identified. Of these, 1/39 (<3%) did not mention consent for adolescents or minors. Thirty-eight policies described consent for HTS and maturity exceptions and were included in the analysis: 10 policies did not provide definition of a minor, one did not use age-based criteria for consent and 27 policies detailed age-based of consent criteria (see online supplemental file 3, included policies). Among included policies, eight limited HTS access without parental consent to persons ≥18 years of age, while 19 (50%) permitted HTS for adolescents <18 years old without parental consent. Of these, six allowed HTS at ≥12 years old, two at ≥13, two at ≥14, five at ≥15 and four at ≥16 (table 2).

The minimum age of consent was 12 years, reported in six countries, most of which were high burden (Lesotho, Mozambique, Rwanda, South Africa, Eswatini and Uganda) (table 2).

Among high-burden countries, 80% (n=8/10) allowed younger adolescents to access testing independent of parents/guardians (Botswana, NA; Eswatini, 12; Lesotho, 12; Malawi, 13; Mozambique, 1; Namibia, 14; South Africa, 12; and Uganda, 12), while 20% (n=2/10) limited independent consent to older adolescents (Zambia, 16, and Zimbabwe, 16) (table 3).

**Table 2** Age of consent to HTS policies in sub-Saharan Africa

| National policies reviewed (n=49) (categories not mutually exclusive) | No. of countries (%) |
|---|---|
| Policy existing for HTS, age of consent clearly defined (including no age limit) | 30 (61.2%) |
| Policy existing for HTS but no age of consent specified* | 8 (16.3%) |
| *Total number of countries with policies discussing age of consent to HTS* | *38 (77.6%)* |
| Policy existing for HTS but no mention of 'adolescent' or 'minor' | 1 (2.0%) |
| No policy for HTS | 10 (20.4%) |
| *Total number of countries either without HTS policies or that have policies that do not mention adolescents or minors* | *11 (22.4%)* |
| **National age of consent to HTS (n=38)**<br>**The year following the country indicates the year of publication for national policy.** | **No. of countries (%)** |
| **Age of minor not defined**<br>Angola, 2015[34]; Burundi, 2014[35]; Cameroon, 2015[36]; Cape Verde, 2005[37]; Chad, 2011[38]; the Comoros, 2007[39]; DRC, 2017[40]; Guinea, 2018[41]; Nigeria, 2016[42]; Sudan, 2016[43] | 10 (26.3%) |
| **No age limit—all or most adolescents eligible†**<br>Botswana, 2016[44] | 1 (2.6%) |
| **12 years**<br>Lesotho, 2016[45]; Mozambique, 2015[46]; Rwanda, 2016[47]; South Africa, 2016[48]; Eswatini, 2018[49]; Uganda, 2016[50] | 6 (15.8%) |
| **13 years**<br>The Gambia, 2014[51]; Malawi, 2016[52] | 2 (5.3%) |
| **14 years**<br>Liberia, 2015[53]; Namibia, 2016[54] | 2 (5.3%) |
| **15 years**<br>Ethiopia, 2017[55]; Kenya, 2015[56]; Senegal, 2017[57]; Somalia, 2017[58]; United Republic of Tanzania, 2019[59] | 5 (13.2%) |
| **16 years**<br>Côte d'Ivoire, 2019[60]; Ghana, 2017[61]; Zambia, 2016[62]; Zimbabwe, 2014[63] | 4 (10.5%) |
| **18 years**<br>Benin, 2017[64]; Burkina Faso, 2008[65]; Central African Republic, 2010[66]; Eritrea, 2019[67]; Madagascar, 2011[68]; Mali, 2017[69]; Sierra Leone, 2017[70]; South Sudan, 2017[71] | 8 (21.1%) |

*These countries used the term 'minor' in their guidance, but do not specify an age.
†These countries stipulate considerations other than age to allow access to HTS (eg, shows maturity; of reproductive age; or married, pregnant or engaged in behaviour that would put them at risk).
DRC, Democratic Republic of the Congo; HTS, HIV testing services.

All high-burden country policies reviewed permitted adolescents of 16 years old and younger to access HTS without parental consent pointing to alignment with WHO 2013 guidance and movement towards accessible testing services for adolescents where HIV is a major public health threat. NCPI age of consent policy reporting complemented our findings; however, some instances of disagreement were identified (table 4).

### Maturity exceptions

Maturity exceptions that waive parental consent requirements and effectively lower the age of consent for some groups of minors were identified in more than half (n=20/38) of policies that address age of consent, including those policies that did not specify age-based definitions of the age of majority (table 5).

The most common maturity exceptions were pregnancy (n=15/38), demonstrated maturity and understanding of test (n=13/38), marriage (n=11/38), parenthood (n=10/38) and being at risk of contracting HIV due to behaviours or population group (n=9/38). Only six countries (Liberia, Rwanda, Sierra Leone, Somalia, South Sudan and Zambia) include maturity exceptions for adolescents from key populations group, all related to people who sell sex. Only one country (Liberia) included a maturity exception for street children. One country (Botswana) does not use age to determine adolescent access to HIV testing. Instead, the policy emphasised the need for providers to assess client maturity and client risks.

When stratified by subregion, we identified notable differences the age of consent policies between ESA (n=21 countries) and WCA (n=25 countries) subregions. In WCA, for 32% of countries (n=8/25), no HTS policy was identified (Congo, Equatorial Guinea, Gabon, Guinea-Bissau, Mauritania, Niger, São Tomé and Príncipe, Seychelles and Togo). While in ESA, only two countries (Seychelles and Djibouti) had no identified policy.

**Table 3** HIV testing policies identified and age of consent to HTS in sub-Saharan Africa

| | Country | Age of consent to HTS | Sub-region | HIV burden* |
|---|---|---|---|---|
| 1 | Angola | MND[34] | ESA | Low |
| 2 | Benin | 18 (Age 15 if adolescent shows maturity)[64] | WCA | Low |
| 3 | Botswana | NA[44] | ESA | High |
| 4 | Burkina Faso | 18 (MME: from 15 years of age.)[65] | WCA | Low |
| 5 | Burundi | MND[72] | WCA | Low |
| 6 | Cameroon | MND (MME: HTS from age 10–13 with parental consent and HTS from age 14–19 if sexually active or family head)[36] | WCA | Low |
| 7 | Cape Verde | MND[37] | WCA | Low |
| 8 | Central African Republic | 18 (MME: if there is no parent or guardian and a child is able to manage the decision, a minor may access HTS independently)[66] | WCA | Low |
| 9 | Chad | MND[38] | WCA | Low |
| 10 | Comoros | MND[39] | ESA | Low |
| 11 | Congo | NI | WCA | Low |
| 12 | Côte d'Ivoire | 16[73] | WCA | Low |
| 13 | Democratic Republic of the Congo | MND[40] | WCA | Low |
| 14 | Djibouti | NI | ESA | Low |
| 15 | Equatorial Guinea | NI | WCA | High |
| 16 | Eritrea | 18[67] | ESA | Low |
| 17 | Eswatini | 12 (MME: people 16 years old or more may access HIVST)[49] | ESA | High |
| 18 | Ethiopia | 15 (MME: mature minors 13–15 years old may consent to HIV testing; those less than 15 years of age cannot consent to HIV testing)[55] | ESA | Low |
| 19 | Gabon | NI | WCA | Low |
| 20 | Gambia | 13 (MME: for those below age 13, parental consent is required unless it is an emergency)[51] | WCA | Low |
| 21 | Ghana | 16 (MME: any person who is between 14 and 16 years old and is sexually active, married, pregnant and a parent or requests HTS is considered to be a mature minor and is able to give full informed consent)[61] | WCA | Low |
| 22 | Guinea | MND[41] | WCA | Low |
| 23 | Guinea-Bissau | NI | WCA | Low |
| 24 | Kenya | 15 (MME: emancipated minor exception; survivors of sexual violence who are under the age of 15 may access HIV testing without parental consent)[56] | ESA | Low |
| 25 | Lesotho | 12[45] | ESA | High |
| 26 | Liberia | 14 (MME: any child who is married and pregnant, works as a commercial sex worker and is a street teenager, family head or with a history of sexual intercourse is regarded as a mature or emancipated minor and can consent for HIV testing)[53] | WCA | Low |
| 27 | Madagascar | 18[68] | ESA | Low |
| 28 | Malawi | 13 (MME: minors of less than 13 years of age may consent to HTS)[52] | ESA | High |
| 29 | Mali | 18[69] | WCA | Low |
| 30 | Mauritania | NI | WCA | Low |
| 31 | Mauritius | NM[74] | ESA | Low |

Continued

| | Country | Age of consent to HTS | Sub-region | HIV burden* |
|---|---|---|---|---|
| 32 | Mozambique | 12 (MME: adolescents married, living as if married, sexually active and head of household may give consent for HIV testing)[46] | ESA | High |
| 33 | Namibia | 14[54] | ESA | High |
| 34 | Niger | NI | WCA | Low |
| 35 | Nigeria | MND[42] | WCA | Low |
| 36 | Rwanda | 12[47] | ESA | Low |
| 37 | São Tomé and Príncipe | NI | WCA | Low |
| 38 | Senegal | 15[57] | WCA | Low |
| 39 | Seychelles | NI | ESA | Low |
| 40 | Sierra Leone | 18 (MME: child survivors of sexual assault or abuse, married young people under age 18, pregnant, parents, minors engaged in behaviour that puts them at risk and child sex workers)[70] | WCA | Low |
| 41 | Somalia | 15 (MME)[58] | ESA | Low |
| 42 | South Africa | 12 (for HIVST, children between age 12 and 18 years should use directly assisted testing options)[48] | ESA | High |
| 43 | South Sudan | 18 (MME: married, pregnant, parents and emancipated)[71] | ESA | Low |
| 44 | Sudan | MND (MME: adolescent KP†)[43] | ESA | Low |
| 45 | Togo | NI | WCA | Low |
| 46 | Uganda | 12[50] | ESA | High |
| 47 | United Republic of Tanzania | 15[59 75] | ESA | Low |
| 48 | Zambia | 16[62] | ESA | High |
| 49 | Zimbabwe | 16 (MME: shows maturity, at risk of HIV, symptomatic and pregnant)[63] | ESA | High |

*Countries were classified as low and high HIV burden using an HIV prevalence threshold of 5%. Data were obtained from Global AIDS Monitoring, 2020.[1]
†KP, key populations
ESA, Eastern and Southern Africa; HIVST, HIV self-testing; HTS, HIV testing services; MME, mature minor exceptions; MND, minor not defined; NA, no age of consent to HTS described in policy; NI, no information; NL, no age of consent to sex law; NM, no mention; WCA, Western and Central Africa.

Of the 17 WCA countries for which a policy was identified, 70% (n=12/17) included clearly defined age of consent for HTS, and five countries (Benin, Burkina Faso, Central African Republic, Mali and Sierra Leone) required parental consent for adolescents under 18 years of age to test. In ESA, there were six countries with an age of consent of 12 years, whereas none were identified in WCA.

### Changes in age of consent policies for HIV testing over time
Thirty-one of the country policies included (table 2) were published after the 2013 release of WHO guidance on HIV and adolescents, which called on policy-makers to detail clear and specific age-based consent language permitting adolescent access to HTS.[23] In 2013, clear age-based criteria for consent to HTS were present in 70% (n=28/40) of the laws and policies reviewed; we identified a similar proportion of policies with clear age-based consent language in 2019, 73% (n=28/38).[7] We observed an increase in the proportion of policies with age-based consent criteria, which permitted adolescents of 16 years or younger to access HTS without parental consent: in 2013, 50% (n=14/28) allowed access without parental consent to those of 16 years or younger, and in 2019, 68% (n=19/28) allowed access without parental consent to those of 16 years or younger. While a majority of policies had been updated since 2013, reduction in age of consent to HTS occurred in just six policies, three high-burden countries (Eswatini, Mozambique and Namibia) and just four low-burden countries (Côte d'Ivoire, the Gambia, Rwanda and United Republic of Tanzania).

### HIV self-testing
No policies in the 2013 review included information on age of consent for HIVST. The first WHO recommendation on HIVST was introduced in 2016. In 2019, nearly half of (44%, 17/38) national policies included self-testing, and 53% of these (n=9/17) clearly defined age

**Table 4** Age of consent to HTS in sub-Saharan Africa national policies (2013 and 2019) and the National Commitments and Policy Instrument (NCPI)

| Age of consent policy | WHO 2013* (n = 33) | | WHO 2019† (n = 38) | | NCPI‡ (n= 49) | |
|---|---|---|---|---|---|---|
| | No of countries (%) | Name of countries | No of countries (%) | Name of countries | No of countries (%) | Name of countries |
| No information/no policy identified (NCPI: no data) | 8 | Chad, the Comoros, Equatorial Guinea, Eritrea, Gabon, the Gambia, São Tomé and Principe, South Sudan | 10 | Congo, Djibouti, Equatorial Guinea, Gabon, Guinea-Bissau, Mauritania, Niger, São Tomé and Principe, Seychelles, Togo | 7 (14.2%) | Cabo Verde, Cameroon, Chad, the Comoros, Eritrea, Guinea-Bissau, São Tomé and Principe, Somalia |
| Policy existing for HTS but no mention of 'adolescent' or 'minor' | 1 (2.1%) | Mauritania | 1 (2.6%) | Mauritius[74] | – | – |
| Age of minor not defined | 7 (17.5%) | Angola, Benin, Burundi, Guinea-Bissau, Madagascar, Mali, Togo | 10 (26.3%) | Angola,[34] Burundi,[35] Cameroon,[36] Cape Verde, Chad,[38] the Comoros,[39] Democratic Republic of the Congo (DRC),[40] Guinea,[41] Nigeria,[42] Sudan[43] | – | – |
| No age limit—all or most adolescents eligible | 5 (12.5%) | Botswana, Cape Verde, Kenya, Mauritius, Somalia (Somaliland) | 1 (2.6%) | Botswana[44] | – | – |
| 12 years (NCPI: yes, for adolescents younger than 12 years) | 3 (7.5%) | Lesotho, South Africa, Uganda | 6 (15.8%) | Lesotho,[76] Mozambique,[46] Rwanda,[47] South Africa,[48] Eswatini,[49] Uganda[50] | 1 (2.0%) | Rwanda |
| 13 years | 1 (2.5%) | Malawi | 2 (5.3%) | The Gambia,[51] Malawi[52] | 0 | – |
| 14 years (NCPI: yes, for adolescents younger than 14 years) | 1 (2.5%) | Liberia | 2 (5.3%) | Liberia,[53] Namibia[54] | 15 30.6% | Benin, Central African Republic, Congo, Gabon, the Gambia, Lesotho, Liberia, Malawi, Mozambique, Namibia, Senegal, South Africa, Togo, Uganda |
| 15 years | 3 (7.5%) | Ethiopia, Rwanda, Senegal | 5 (13.2%) | Ethiopia,[55] Kenya,[56] Senegal,[57] Somalia,[58] United Republic of Tanzania[59] | 0 | – |
| 16 years (NCPI: yes, for adolescents younger than 16 years) | 6 (15.0%) | Congo, Mozambique, Namibia, Swaziland, Zambia, Zimbabwe | 4 (10.5%) | Côte d'Ivoire,[73] Ghana,[61] Zambia,[62] Zimbabwe[63] | 9 (18.3%) | Botswana, Burundi, Cote d'Ivoire, Djibouti, Equatorial Guinea, Guinea, Niger, Zambia, Zimbabwe |

**Table 4** Continued

| Age of consent policy | WHO 2013* (n = 33) | | WHO 2019† (n = 38) | | NCPI‡ (n= 49) | |
|---|---|---|---|---|---|---|
| | Name of countries | No of countries (%) | Name of countries | No of countries (%) | Name of countries | No of countries (%) |
| 18 years (NCPI: yes for adolescents less than 18 years) | Burkina Faso, Cameroon, Central African Republic, Côte d'Ivoire, DRC, Djibouti, Ghana, Guinea, Niger, Nigeria, Seychelles, Sierra Leone, Sudan, United Republic of Tanzania | 14 (35.0%) | Benin,[64] Burkina Faso,[65] Central African Republic,[66] Eritrea,[67] Madagascar,[68] Mali,[69] Sierra Leone,[70] South Sudan[71] | 8 (21.1%) | Angola, Burkina Faso, DRC, Eswatini, Ethiopia, Ghana, Kenya, Madagascar, Mali, Mauritius, Nigeria, Seychelles, Sierra Leone, South Sudan, Sudan, United Republic of Tanzania | 17 (35.6%) |

*2013 review included both policies and laws, while this review included only policies.[7]
†Findings identified in the context of this policy review.
‡Data reported in the UNAIDS National Commitments and Policy Instrument (2017, 2018 and 2019).
HTS, HIV testing services.

of consent for access to HIVST. Among HIVST policies, nine clearly defined age of consent for HIVST (Côte d'Ivoire, 16; Eritrea, 18; Eswatini, 16; Ghana, 16; Mali, 18; Somalia, 18; South Africa, 12 with directly assisted HIVST; South Sudan, 18; and Tanzania, 18). Nine policies (Benin, Cameroon, Democratic Republic of the Congo, Kenya, Lesotho, Malawi, Nigeria, Zambia and Zimbabwe) addressed HIVST but did not include clear definition of age of consent, and among these were two standalone HIVST policies (Zambia and Zimbabwe). HIVST policies were diverse in format; some were standalone, while others integrated HIVST into broader testing service policies. Two policies (Kenya and South Africa) explicitly enabled self-testing but only with direct assistance for those under the age of 18 years, thus distinguishing HIVST from conventional provider-delivered options. Eswatini, United Republic of Tanzania and Somalia had a higher age of consent for HIVST (16 years for Eswatini and 18 years for Somalia and United Republic of Tanzania) than for provider-delivered methods (12 years for Eswatini and 15 years for United Republic of Tanzania and Somalia).

### NCPI

NCPI, a mechanism for monitoring national laws and policies, captures age of consent regulations for HIV testing and other reproductive health services among other themes.[28 30] Our review findings complement the policy reporting in NCPI but do not capture consistent country-level information on age of consent to HIV testing (table 3). The discrepancy of our findings with those reported by NCPI highlights challenges in aggregating policy information in surveys and research. Since NCPI includes age of consent policy information covering HTS, ART and sexual and reproductive health and rights (SRHR) services, it provides regular regional intelligence and offers a platform to routinely review, validate and engage around age of consent policies.

### DISCUSSION

Our review found that half of policies in SSA allowed adolescents aged 16 years and younger to provide independent consent for HIV testing.

We found the minimum specified age of consent for HTS in any SSA country was 12 years with one country reporting no age-based criteria. In comparison, the minimum specified age of consent for HTS in Western Europe and North America is 14 years, with some countries reporting no age-based criteria and many countries not reporting.[28] There has been substantial progress in SSA since 2013 when WHO encouraged countries to review and adapt consent policies to make HIV testing accessible to adolescents, with 31 updated policies identified. Among these, seven lowered the age of consent.[23] Policy-makers in SSA have moved to address HIVST, but there is more work to be done in both developing policy and including clearly defined age of consent language.

| Table 5 | Exceptions for HTS below age of consent | |
|---|---|---|
| **Exceptions for testing below stated legal age of consent** **The age in parentheses indicates the national age of consent to HTS before the exception is applied** | | **No of countries** |
| **Shows maturity and understanding of the process and potential results (healthcare provider discretion)** Benin (18 years),[64] Botswana (no age limit),[44] Burkina Faso (18 years but 'mature minor' exception from 15 years),[65] Ethiopia (15 years but 'mature minor' exception from 13 years),[55] Ghana (16 years),[61] Liberia (14 years),[53] Malawi (13 years),[53] Mozambique (12 years),[46] Namibia (14 years),[54] Sierra Leone (18 years),[70] Somalia (15 years),[58] South Africa (12 years),[48] South Sudan (18 years),[71] Zimbabwe (16 years)[77] | | 13 |
| **At risk of contracting HIV (eg, sexually active)** Burkina Faso (18 years but 'mature minor' exception from 15 years),[65] Liberia (14 years),[53] Namibia (14 years),[54] Sierra Leone (18 years),[70] Somalia (15 years),[58] Sudan (age of 'minor' not defined),[43] United Republic of Tanzania (15 years),[75] Zimbabwe (16 years)[77] | | 9 |
| **Symptomatic** Benin (18 years),[64] Liberia (14 years),[53] Zimbabwe (16 years)[77] | | 3 |
| **Pregnant** Benin (18 years),[64] Botswana (no age limit),[44] Burkina Faso (18 years, 'mature minor' exception from 15 years),[65] Cameroon (age of minor not defined),[36] Ghana (16 years),[61] Liberia (14 years),[53] Mozambique (12 years),[46] Namibia (14 years),[54] Rwanda (12 years),[47] Sierra Leone (18 years),[70] Somalia (15 years),[58] South Sudan (18 years),[71] United Republic of Tanzania (15 years),[75] Zambia (16 years),[62] Zimbabwe (16 years)[63] | | 15 |
| **Parent (adolescent is already a parent)** Benin (18 years),[64] Botswana (no age limit),[44] Cameroon (age of minor not defined),[36] Ghana (16 years),[61] Mozambique (12 years),[46] Namibia (14 years),[54] Sierra Leone (18 years),[70] Somalia (15 years),[58] South Sudan (18 years),[71] Zambia (16 years)[62] | | 10 |
| **Head of household** Central African Republic (18 years),[66] Liberia (14 years),[53] Mozambique (12 years),[46] Zambia (16 years)[62] | | 4 |
| **Married** Benin (18 years),[64] Botswana (no age limit),[44] Burkina Faso (18 years, 'mature minor' exception from 15 years),[65] Ghana (16 years),[61] Liberia (14 years),[53] Mozambique (12 years),[46] Namibia (14 years),[37] Sierra Leone (18 years),[70] South Sudan (18 years),[71] United Republic of Tanzania (15 years)[59 75] Zambia (16 years)[62] | | 11 |
| **Sex workers** Liberia (14 years),[53] Rwanda (12 years),[47] Sierra Leone (18 years),[70] Somalia (15 years),[58] South Sudan (18 years),[71] Zambia (16 years)[62] | | 6 |
| **Street children** Liberia (14 years)[53] | | 1 |
| **Emancipated minor** Botswana (no age limit),[44] Central African Republic (18 years),[66] Kenya (15 years),[56] Liberia (14 years),[53] Madagascar (18 years),[68] South Sudan (18 years)[71] | | 6 |

Our review also identified gaps in age of consent policies in SSA, particularly WCA: overall, about a quarter of policies (n=10/38) did not clearly describe age of consent, and 21% restricted HTS to those 18 years or above.

In this review, we assessed only HIV testing policies. However, policies should facilitate integration and linkage to promote comprehensive adolescent health packages that deliver the benefits of HIV prevention, treatment and other SRHR services so that public health impact can be realised. In countries with clear age of consent policies (including many in ESA), alignment of HTS policies with other SRHR and HIV prevention policies should be prioritised. In countries without policies, or in those with vague policies (including many in WCA), development of guidance with clear consent language that makes HIV testing, SRHR and HIV services broadly available to adolescents is needed. In all settings, key population and sexually active adolescents need policies that allow them to provide their own consent for HTS.[28] One recent country example of

policy change took place in United Republic of Tanzania, where, in 2019 following a period of coordinated political advocacy, parliamentary support for lowering age of consent for HIV testing from 18 to 15 years was obtained; age of consent for HIVST remained at 18 years of age.

A highlight of this review is the assessment of policies around HIVST, a highly acceptable way to engage adolescents in HIV testing and to make testing accessible for adolescents and young people.[31] In 2013, only two countries (Kenya and the USA) had HIVST policies, and there were no WHO recommendations regarding their use. Further, the US Food and Drug Administration restricted distribution by requiring proof of age for those accessing HIVST and limiting use to those of 17 years and older although initial studies included those of 14 years and older in subgroup analysis.[32]

Following the release of the first WHO guidance on self-testing for HIV in 2016, our 2019 review identified 17 country policies that addressed HIVST. Adolescents who

self-test need tools and resources to support linkage to prevention and treatment services, and approaches that deliver self-testing to this population through assisted and unassisted methods have been demonstrated effective and acceptable in SSA.[31] Two countries (South Africa and Kenya) stipulated assisted HIVST approaches for younger adolescents, thus making the modality available while ensuring a minimum standard of support for performing the self-test, interpreting the result and linking to other services.

Policies frequently included maturity exceptions that eliminate parental consent requirements for adolescents seeking HTS, including those from key populations who are at particularly high risk of HIV and may also face social and structural barriers to testing, prevention and care services.[33]

Many other vulnerable adolescents are left out and overlooked: only six countries (Liberia, Rwanda, Sierra Leone, Somalia, South Sudan and Zambia) have policies that address adolescents who sell sex, and exceptions for adolescents from other key populations were not identified. Exceptions did frequently address adolescent girls; those for marriage, pregnancy, involvement in sex work and sexual activity may effectively lower age of consent for some girls; however, girls may not feel empowered to seek health services or to take a test when services are not designed to be accessible to adolescents. Overall, we found great variation in maturity exceptions and are not able to ascertain how these are interpreted and put into practices by implementers and service providers or how they ultimately impact access to HIV testing for adolescents. In settings where maturity exception policies are present, programmes should assess fidelity as well as seek and consider adolescent perspectives while they aim to strengthen service delivery for HIV testing and prevention.

### Strengths and limitations

Documents from nearly forty countries were reviewed using a standard approach to extraction and analysis. Despite a robust approach, some policy documents may not have been identified. Due to varied document formats, the reviewers employed a non-automatised data extraction that was complemented by using key words to maintain repeatability. This review included national policy documents but did not assess national laws, which contribute to a potentially limiting operational context to better understand guidance, which informs routine implementation.[9] Since the review did not assess implementation, it does not provide insight into how policies ultimately impact HTS delivery and access for adolescents.

Further policy reviews and national assessments that address alignment on age of consent within and between HIV testing, prevention and treatment and sexually transmitted infection (STI) and other sexual and reproductive health services will illuminate opportunities to strengthen policy and implementation. Finally, while a significant proportion of policies reviewed did not define the age of consent, which limited assessment of regional trends, this did not hinder our ability to assess whether national policies contained clear description of age of consent to test.

### CONCLUSIONS

Since 2013, 31 SSA countries have updated policies that include age of consent to HTS, with seven reducing the age of consent to 12 years or eliminating age-based criteria. Our findings suggest that adolescents in 19/38 (50%) of SSA countries may access HTS without consent of a parent or guardian at varying ages: 16, 15, 14 or 12 years. Access to HTS for adolescents remains limited in WCA, where policies are fewer and older and have vague or no age of consent language. Adolescence, a period of rapid physical, sexual and cognitive development, is a time when young people need health services that address their diverse and evolving needs. HTS policies should acknowledge the right of adolescents to provide their own consent to testing, particularly through newly introduced approaches like HIVST, and be aligned with national SRHR guidance so that adolescents can receive the benefits of services for contraception, voluntary medical male circumcision, STIs, HIV prevention and antiretroviral treatment. Where adolescents have supportive parents or trusted older relatives, involving them in testing is often of great benefit to coping with a result and discussing linkage to other services post testing. As countries renew efforts to ensure that 95% of people with HIV know their status and to improve testing, prevention and treatment access for adolescents, lowering or eliminating age of consent for HIV testing should continue to be prioritised.

**Correction notice** This article has been corrected since it was published Online First. The funding statement has been updated.

**Acknowledgements** The authors thank all country and regional contacts for policies for the review.

**Contributors** CJ and RB devised and supervised the review. MB-D and CJ designed the protocol. MB-D conducted the search, and MT conducted the screening and data extraction. MB-D and MT qualitatively synthesised the findings and conducted descriptive analyses. MB-D and LH drafted the manuscript with input from MT, CJ and MSJ. MB-D, MT, LH, BJ, CQ, WA, MSJ, KS, RB and CJ reviewed the draft, provided critical review and read and approved the final manuscript.

**Funding** Funding for this review was provided by the US President's Emergency Plan for AIDS Relief (PEPFAR) through World Health Organization and USAID's Consolidated WHO Grant, Agreements US-2015-0839 and US-2016-940. This work was supported, in whole or in part, by the Bill & Melinda Gates Foundation.

**Competing interests** None declared.

**Patient consent for publication** Not required.

**Provenance and peer review** Not commissioned; externally peer reviewed.

**Data availability statement** Data are available upon reasonable request. Data may be obtained from a third party, and some data are publicly available. Most policies included in this review are available through the following websites: (1) https://aidsfree.usaid.gov/resources/guidancedata/hts and (2) http://www.hivpolicywatch.org/database.html. If information on a policy cannot be found through these sources, please contact the authors of this review for additional information.

**ORCID iD**
Magdalena Barr-DiChiara http://orcid.org/0000-0002-7404-7843

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
