## [Reviewer comments · BMJ Open]

ARTICLE DETAILS

TITLE (PROVISIONAL)	Adolescents and age of consent to HIV testing: an updated review of national policies in sub-Saharan Africa
AUTHORS	Barr-DiChiara, Magdalena; Tembo, Mandikudza; Harrison, Lisa; Quinn, Caitlin; Ameyan, Wole; Sabin, Keith; Jani, Bhavin; Jamil, Muhammad; Baggaley, R; Johnson, Cheryl

VERSION 1 – REVIEW

REVIEWER	Kanagasabai, Udhayashankar Centers for Disease Control and Prevention Center for Global Health, Division of Global HIV and TB
REVIEW RETURNED	10-May-2021

GENERAL COMMENTS	This is very well written paper that highlights and important issue.
--

REVIEWER	Wariri, Oghenebrume MRC Laboratories The Gambia, Vaccines and Immunity Theme
REVIEW RETURNED	16-May-2021

GENERAL COMMENTS	Thank you for inviting me to review this important manuscript. In the manuscript, the authors evaluated country progress and policies on the age of consent for HIV testing in line with the 2013 recommendation of the WHO for countries to review policies on the age of consent for HIV testing services. They provide an updated review focussing on countries in sub-Saharan Africa, which in my opinion, provides useful baseline data to support country-level HIV testing policy changes in SSA. I give tremendous credit to the authors of this manuscript because it is very clear that a lot of work has gone into preparing this document. With that being said, there are two major comments and other minor issues which the authors should consider addressing to improve the overall quality of the manuscript Major: 1. It is not immediately apparent why national HTS policies in force in SSA from 2005-2019 were included in the analysis instead of those from 2013-2019. The authors clearly made reference to the 2013 WHO guidance on HIV and adolescents which encouraged countries to consider revising parental consent requirements. Based on this reference, readers' expectation would be that the authors were tracking policy changes in SSA since that recommendation in 2013 upwards and not policies before 2013. In fact, in the concluding statement of the introduction, the authors state that " Here we review policies in force following the 2013 WHO guidelines to evaluate country progress and uptake of WHO
---

	guidance on the age of consent for HIV testing, including the age of consent for accessing HIVST” 2. In the methods section, the authors should consider developing a summary table (figure) that states the various WHO repositories, government websites, and non-governmental websites with the specific dates that they were searched. This is important because it improves transparency and reproducibility for other researchers who might consider updating this review in the coming years. Minor: 3. On page 4, lines 21-26, the sentence referring to the projected number of young people in 2055 should be referenced. 4. On page 4, lines 34-39, in the sentence “In eastern and southern Africa (ESA) only 27% of adolescent girls and 16% of adolescent boys aged 15-19 received an HIV test in the last year and lower uptake was observed in western and central Africa(WCA)”, a specific year should be stated instead of using “in the last year”. 5. On page 5, lines 43-44, the sentence starting with “Recent findings show that a when...” should read, “Recent finding show that when....” 6. On page 6, line 13, “non-governmental” is repeated twice. Please delete one of them 7. On page 6, line 57, should read “Two authors (MBD and MT) extracted information from policy documents independently”
--	--

REVIEWER	Rusley, Jack Brown University, Division of Adolescent Medicine
REVIEW RETURNED	19-May-2021

GENERAL COMMENTS	Abstract: Consider mentioning why only 38 of 40 policies were included (I believe you do later in the paper, but would include in abstract as well). Consider rephrasing results sentence beginning with "19/38 (50%) permitted..." since it makes it seem as though these 50% permit testing without consent for all youth ages 12-16, even though the next sentence makes it clear that countries actually vary with what age the consent policy takes effect. Page 8 lines 15-16 - I think this should say "high burden countries" not "high burden policies"? Page 9 lines 21-29 - I think this should say "increase" instead of "decrease"? Page 13 lines 25-27: Not sure what "reach the first 95" means - consider clarifying. In this same sentence, would consider clarifying "outcomes for adolescents" - I assume you are referring to health outcomes, but perhaps helpful to be clear which ones might benefit from increased access to testing. Limitations: Consider adding language discussing how many countries in SSA are not member states of the WHO, and why they were not included. Also, I was struck by the absence of comment on what the lowest age of self-consent should be - while I recognize this is a complex issue and perhaps a specific number is not actually the best approach, I do think it would be valuable for the authors to comment on this (especially since I would imagine policy-makers would wonder "what should the lower age be?" and would appreciate guidance on how to think about this question).
--

VERSION 1 – AUTHOR RESPONSE

Reviewer Reports:

Reviewer: 1

Dr. Udhayashankar Kanagasabai, Centers for Disease Control and Prevention Center for Global Health

Comments to the Author:

This is very well written paper that highlights and important issue.

Reviewer: 2

Dr. Oghenebrume Wariri, MRC Laboratories The Gambia

Comments to the Author:

Thank you for inviting me to review this important manuscript.

In the manuscript, the authors evaluated country progress and policies on the age of consent for HIV testing in line with the 2013 recommendation of the WHO for countries to review policies on the age of consent for HIV testing services. They provide an updated review focussing on countries in sub-Saharan Africa, which in my opinion, provides useful baseline data to support country-level HIV testing policy changes in SSA.

I give tremendous credit to the authors of this manuscript because it is very clear that a lot of work has gone into preparing this document.

With that being said, there are two major comments and other minor issues which the authors should consider addressing to improve the overall quality of the manuscript

Major:

1. It is not immediately apparent why national HTS policies in force in SSA from 2005-2019 were included in the analysis instead of those from 2013-2019. The authors clearly made reference to the 2013 WHO guidance on HIV and adolescents which encouraged countries to consider revising parental consent requirements. Based on this reference, readers' expectation would be that the authors were tracking policy changes in SSA since that recommendation in 2013 upwards and not policies before 2013. In fact, in the concluding statement of the introduction, the authors state that "Here we review policies in force following the 2013 WHO guidelines to evaluate country progress and uptake of WHO guidance on the age of consent for HIV testing, including the age of consent for accessing HIVST"

Response: Noting the lack of clarity in descriptions of methods and specifically inclusion criteria a revision has been included. The reason policies prior to 2013 were included in the analysis was that some countries had not updated policies and it was determined that these had older policies in force indicating no change or update in national policy following the 2013 guidance from WHO that countries review and update policies for age of consent to HTS.

2. In the methods section, the authors should consider developing a summary table (figure) that states the various WHO repositories, government websites, and non-governmental websites with the specific dates that they were searched. This is important because it improves transparency and reproducibility for other researchers who might consider updating this review in the coming years.

Response: Detailed information on the search strategy and process has been provided in the revised supplementary file. An additional note has been added describing that the repository is maintained by and available from WHO. We welcome feedback on the adequacy of this response.

Minor:

3. On page 4, lines 21-26, the sentence referring to the projected number of young people in 2055 should be referenced.

Response. The reference has been added.

4. On page 4, lines 34-39, in the sentence "In eastern and southern Africa (ESA) only 27% of adolescent girls and 16% of adolescent boys aged 15-19 received an HIV test in the last year and lower uptake was observed in western and central Africa(WCA)", a specific year should be stated instead of using "in the last year".

Response. Year has now been included.

5. On page 5, lines 43-44, the sentence starting with "Recent findings show that a when..." should read, "Recent finding show that when...."

Response. The typo has been resolved.

6. On page 6, line 13, "non-governmental" is repeated twice. Please delete one of them Response. The typo has been resolved.

7. On page 6, line 57, should read "Two authors (MBD and MT) extracted information from policy documents independently" Response. The suggestion has been incorporated.

Reviewer: 3

Dr. Jack Rusley, Brown University

Comments to the Author:

Abstract: Consider mentioning why only 38 of 40 policies were included (I believe you do later in the paper, but would include in abstract as well).

Response. A rewording has been proposed to address.

Consider rephrasing results sentence beginning with "19/38 (50%) permitted..." since it makes it seem as though these 50% permit testing without consent for all youth ages 12-16, even though the next sentence makes it clear that countries actually vary with what age the consent policy takes effect.

Response. Reworded to make more clear.

Page 8 lines 15-16 - I think this should say "high burden countries" not "high burden policies"?

Response. Reworded.

Page 9 lines 21-29 - I think this should say "increase" instead of "decrease"? Response. Resolved.

Page 13 lines 25-27: Not sure what "reach the first 95" means - consider clarifying. In this same sentence, would consider clarifying "outcomes for adolescents" - I assume you are referring to health outcomes, but perhaps helpful to be clear which ones might benefit from increased access to testing.

Response. Resolved.

Limitations: Consider adding language discussing how many countries in SSA are not member states of the WHO, and why they were not included.

Response. Countries were not excluded from the review based on WHO membership. Country exclusion from the review was determined by absence of an identified policy which that inclusion criteria.

Also, I was struck by the absence of comment on what the lowest age of self-consent should be - while I recognize this is a complex issue and perhaps a specific number is not actually the best approach, I do think it would be valuable for the authors to comment on this (especially since I would imagine policy-makers would wonder "what should the lower age be?" and would appreciate guidance

on how to think about this question).

Response: While the authors have not included a minimum age of consent recommendation. We have included two revisions in the conclusion paragraph. These are:

Adolescence is a time of rapid physical, sexual and cognitive development is a time when young people need health services that address their diverse and evolving needs.

As countries renew efforts to ensure that 95% of people with HIV know their status, and to improve testing, prevention and treatment access for adolescents, lowering or eliminating age of consent for HIV testing should continue to be prioritized.

Reviewer: 1

Competing interests of Reviewer: I have no competing interest.

Reviewer: 2

Competing interests of Reviewer: None

Reviewer: 3

Competing interests of Reviewer: None

Note: The co-authors note that the manuscript is over the recommended word count but has been approved by the review team without a suggestion to reduce length.

In response to the comments from the editorial team the following additions were made:

1. Supplementary files these have been re-uploaded in PDF format.
2. The affiliation for Mandi Tembo has been updated to Department of Infectious Disease Epidemiology, London School of Hygiene and Tropical Medicine.
3. The initials of all authors now appear in the contributor statement.

CJ and RB devised the review. MBD and CJ designed the protocol. MBD conducted the search, and MT conducted the screening and data extraction. CJ and RB supervised the review.

MBD and MT qualitatively synthesized the findings and conducted descriptive analyses. MBD and LH drafted the manuscript with input from MT, CJ and MJ. MBD, MT, LH, BJ, CQ, WA, MJ, KS, RB, CJ reviewed the draft, provided critical review, and read and approved the final manuscript.

Thank you for your assistance and review.

VERSION 2 – REVIEW

REVIEWER	Wariri, Oghenebrume MRC Laboratories The Gambia, Vaccines and Immunity Theme
REVIEW RETURNED	27-Jun-2021
GENERAL COMMENTS	The authors have appropriately addressed the issues I raised in the previous round of review. I have no further comments. I wish them well in their future research endeavors.